# Effectiveness of Dietitian-Involved Lifestyle Interventions in Diabetes Management Among Arab Populations: A Systematic Review and Meta-Analysis

**DOI:** 10.3390/nu16244283

**Published:** 2024-12-11

**Authors:** Galia Sheffer-Hilel, Omaima Abd Elqader, Layla Suliman, Einav Srulovici

**Affiliations:** 1The Cheryl Spencer Department of Nursing, Faculty of Social Welfare and Health Sciences, Haifa University, Haifa 3498838, Israel; omaima.3f@gmail.com (O.A.E.); lt30-5@hotmail.com (L.S.); esrulovici@univ.haifa.ac.il (E.S.); 2Nutrition Sciences Department, Faculty of Sciences, Tel-Hai Academic College, Qiryat Shemona 1220800, Israel

**Keywords:** diabetes mellitus, Arabs, lifestyle interventions, dietitian

## Abstract

Background/Objectives: Diabetes prevalence is high among Arab populations, where cultural practices present barriers to effective glycemic control. Despite guidelines recommending the involvement of dietitians in diabetes management, evidence of the effectiveness of dietitian-involved interventions in these populations remains limited. This systematic review and meta-analysis evaluated the effectiveness of dietitian-involved lifestyle interventions among Arab populations with prediabetes or diabetes. Methods: PubMed, CINAHL, and the Cochrane Library were systematically searched for studies employing experimental and quasi-experimental designs with interventions involving dietitians. All reported outcomes were considered to ensure a comprehensive review. The protocol was registered in PROSPERO (registration number CRD42024555668). Results: The meta-analysis showed significant reductions in glycosylated hemoglobin (HbA1c) levels (−0.41; 95% CI: −0.67, −0.16), body mass index (BMI) (−0.28; 95% CI: −0.36, −0.19), and increases in high-density lipoprotein cholesterol HDL-C) (0.60; 95% CI: 0.36, 0.85) compared to usual care. Subgroup analysis indicated variations based on follow-up duration. The certainty of the evidence was low due to heterogeneity and potential biases, emphasizing the need for further high-quality research to confirm these findings. Conclusions: Interventions involving dietitians improve glycemic control and metabolic outcomes among Arab populations. Given the unique cultural challenges in this population, culturally tailored and personalized interventions are essential to overcome barriers and improve diabetes-related outcomes. Future studies should focus on expanding the evidence base, integrating these findings into healthcare policies, and exploring strategies for long-term sustainability and broader application.

## 1. Introduction

Diabetes mellitus (diabetes) is a global health concern, affecting approximately 9.8% of adults aged 20–79 years, equating to over half a billion individuals worldwide [1]. This chronic condition is influenced by various behavioral and sociodemographic factors, leading to differing prevalence rates across societies [2]. Diabetes prevalence is notably high in the Arab region, particularly in the Gulf Cooperation Council countries, such as the United Arab Emirates, Bahrain, Qatar, Oman, Saudi Arabia, and Kuwait [3]. According to the 2021 report by the International Diabetes Federation, diabetes incidence rates in these Arab countries range from 6% to 25% [4].

The rising rates of diabetes pose a serious concern for healthcare systems due to the associated macrovascular and microvascular complications. These complications include, but are not limited to, cardiovascular diseases, retinopathy, and nephropathy, which consequently lead to increased premature mortality [5]. As the prevalence of diabetes continues to rise, managing its associated complications becomes increasingly challenging, demanding substantial healthcare resources and consuming a significant portion of healthcare budgets [6]. The economic burden of diabetes includes direct medical costs, such as hospitalizations, medications, and outpatient care, as well as indirect costs related to lost productivity and disability [7]. Therefore, effectively addressing diabetes is crucial to mitigating its impact on both individual health and healthcare systems. Implementing comprehensive diabetes management strategies can not only improve patient outcomes but also reduce healthcare expenditures and enhance the overall quality of life for those affected by diabetes [8,9].

Current diabetes management focuses on regulating blood glucose levels through a combination of dietary modifications, insulin, and/or oral medications utilization, maintaining a healthy body weight, regular physical activity, self-monitoring of blood glucose, smoking cessation counseling, and psychosocial care [10]. However, cultural practices in Arab societies often present significant barriers to achieve effective glycemic control. Social customs such as hosting guests with abundant food offerings can impede individuals’ ability to maintain a healthy diet [11]. In many Arab cultures, hospitality is highly valued, and guests are often presented with large quantities of high-calorie, carbohydrate-rich foods, making it challenging for individuals with diabetes to adhere to dietary recommendations. Additionally, festive occasions such as Ramadan and Eid involve communal meals and sweets that are high in sugar and fats, further complicating blood glucose management [12]. During Ramadan, the fasting period is followed by large meals that can cause significant fluctuations in blood glucose levels, posing risks for individuals with diabetes [13]. A shift from traditional, diverse diets rich in fruits, whole grains, and vegetables to Westernized diets high in refined carbohydrates, trans fats, saturated fats, salt, and sugar has further complicated dietary management in these regions [14]. These cultural and nutritional barriers underscore the importance of developing diabetes management strategies that are both clinically effective and culturally sensitive, ensuring that dietary recommendations are practical and acceptable within the framework of Arab societal norms and traditions.

While most studies on diabetes management and lifestyle interventions have been conducted in Western contexts [15,16], there is a significant lack of research focusing on interventions tailored to the unique cultural practices, dietary habits, and social norms of Arab populations [17]. Given the region’s high diabetes prevalence and the culturally ingrained dietary practices, particularly those related to social customs and religious events, there is a pressing need for interventions that respect and integrate these practices while addressing health needs. Without tailored interventions, standard diabetes management programs may fail to achieve optimal outcomes in these populations.

Dietary modification is a critical component of diabetes management, requiring the expertise of dietitians. Dietitians play a vital role in counseling and supporting patients to achieve optimal dietary patterns for blood glucose control and mitigating associated risk factors. Research indicates that clinical outcomes, such as better regulation of glycaemia and weight loss, are more effectively achieved when dietary interventions involve dietitians compared to other healthcare professionals [18]. Systematic reviews have shown that dietitians improve patient outcomes through individual consultations [19,20] and that inter-professional collaborative practice, including dietitians, delivers improved patient outcomes for diabetes and hypertension [21]. Thus, intervention programs addressing dietary changes should incorporate dietitians to ensure culturally sensitive and effective dietary management. However, there is limited evidence on the effectiveness of dietitian-involved programs specifically among Arab individuals. Research is needed to evaluate the overall effectiveness of dietitian-involved programs in this population, ensuring that they are adapted to meet the unique dietary and cultural needs of Arab patients with diabetes [22].

Evidence indicates that adopting a healthy lifestyle is crucial for both the prevention and management of diabetes [23]. For example, a systematic review incorporating 28 studies demonstrated that lifestyle interventions significantly lowered hemoglobin A1c (HbA1c) levels compared to usual care for patients with diabetes [24]. Despite these positive effects observed in the general population, the impact of lifestyle interventions appears to be less pronounced among Arab populations [25]. This discrepancy highlights the urgent need for tailored intervention programs that address the unique cultural and dietary practices of Arab populations.

Given the pivotal role dietitians play in managing diabetes, it is crucial to evaluate the effectiveness of dietitian-involved programs. While these interventions have been proven effective in Western contexts, there is limited research on their outcomes in Arab populations, where cultural and dietary factors may present unique challenges. Therefore, this systematic review and meta-analysis aims to assess the effectiveness of dietitian-involved lifestyle interventions among Arab patients with prediabetes or diabetes. By addressing this gap, we aim to provide valuable insights into the potential of dietitian-involved programs to deliver culturally sensitive and effective diabetes management.

## 2. Method

### 2.1. Search Methods

The systematic review employed three electronic databases—PubMed, CINAHL, and the Cochrane Library. No publication year limits were set to capture all relevant studies, including older research, given the relatively limited body of work addressing interventions involving dietitians in Arab populations. The search strategy aimed to identify all research studies evaluating interventions designed to promote a healthy lifestyle and improve diabetes outcomes among Arab patients with diabetes. This systematic review followed the Preferred Reporting Items for Systematic Reviews and Meta-Analysis (PRISMA) 2020 guidelines [26], and its protocol was registered in PROSPERO with the registration number CRD42024555668.

Medical Subject Heading (MeSH) terms included the following: (“Diabetes Mellitus”[MeSH Terms]) AND ((“Arabs”[MeSH Terms]) OR (“Algeria”[MeSH Terms]) OR “Bahrain”[MeSH Terms] OR “Comoros”[MeSH Terms] OR “Djibouti”[MeSH Terms] OR “Egypt”[MeSH Terms] OR “Iraq”[MeSH Terms] OR “Jordan”[MeSH Terms] OR “Kuwait”[MeSH Terms] OR “Lebanon”[MeSH Terms] OR “Libya”[MeSH Terms] OR “Mauritania”[MeSH Terms] OR “Morocco”[MeSH Terms] OR “Qatar”[MeSH Terms] OR “Saudi Arabia”[MeSH Terms] OR “Somalia”[MeSH Terms] OR “Sudan”[MeSH Terms] OR “Syria”[MeSH Terms] OR “Tunisia”[MeSH Terms] OR “United Arab Emirates”[MeSH Terms] OR “Yemen”[MeSH Terms])) AND (“Program Evaluation”[MeSH Terms] OR “Health Promotion”[MeSH Terms] OR “Self-Management”[MeSH Terms] OR “Health Behavior”[MeSH Terms] OR “Self-Care”[MeSH Terms] OR “Patient Education as Topic”[MeSH Terms] OR “Evaluation Study”[Publication Type] OR “Evaluation Studies as Topic”[MeSH Terms]).

### 2.2. Inclusion and Exclusion Criteria

The inclusion criteria were as follows: (a) studies had to be peer reviewed journal articles, (b) studies published in either the English or Arabic language were eligible, (c) the intervention was required to include a dietitian as part of the intervention team, (d) studies needed to employ a quasi-experimental or experimental design, and (e) studies had to focus on interventions aimed at improving adherence to diabetes self-care among Arab individuals aged 18 years or older with diabetes or prediabetes. The inclusion of prediabetes and diabetes populations reflects their shared pathophysiological mechanisms, including insulin resistance and chronic inflammation, and acknowledges the potential overlap in dietary and lifestyle interventions for these groups. Articles involving non-Arab populations were included only if they provided separate and distinct analyses specifically focused on Arab participants. Exclusion criteria involved studies collecting data at a single point in time without follow-up, presenting observations on individual cases, analyzing data at the population level rather than individual-level data, summarizing the existing literature without presenting original research findings, and non-research articles expressing personal viewpoints or opinions. These criteria ensured that the included studies provided robust and relevant pre–post outcome data on interventions for diabetes management in Arab populations.

### 2.3. Search Outcomes

The literature review process for the systematic review involved several steps. Initially, the titles and abstracts of the relevant literature were extracted from the three databases, and duplicate records were identified and removed. Subsequently, the titles were screened and those that did not meet the predefined inclusion criteria were excluded. In the next phase, the abstracts of the remaining articles were thoroughly reviewed, and those meeting the inclusion criteria proceeded to full-text evaluation, during which the relevant data were extracted. To ensure rigor and reliability, two independent reviewers screened the titles, abstracts, and full-text articles independently while also extracting data and assessing the quality of the included articles. There were only a few disagreements that required discussion between the reviewers, and these were openly addressed and resolved until a consensus was reached. Finally, a comprehensive manual search of reference lists within the included articles was then conducted to identify additional studies for potential inclusion.

### 2.4. Quality Assessment

The methodological quality of the included studies was assessed using the Downs and Black [27] checklist, a widely recognized tool for evaluating randomized and non-randomized studies. This checklist systematically evaluates measures implemented to mitigate bias and errors in study design, execution, and result analysis. The checklist comprises 27 items that examine aspects such as reporting, external validity, internal validity, and statistical power of clinical trials. Each item was scored on a binary scale of 0 or 1, with the exception of one item that allowed for a score of 0, 1, or 2 to assess the provision of principal confounders (i.e., no, partially, or yes, respectively). Consequently, the maximum attainable score on the adjusted checklist for this review was 28 points.

### 2.5. Data Synthesis and Analysis

The systematic review encompassed all outcomes reported in the included articles. No predefined outcomes were selected beforehand; instead, the review considered all the available information presented in the included studies. A meta-analysis was conducted to assess the effectiveness of the various interventions across different follow-up periods on multiple outcomes. All eligible studies were kept for meta-analysis regardless of their quality score, as relatively scarce research has been conducted in the field. Statistical analyses were performed using R version 4.3.3 statistical software. The pooled treatment effect (TE) sizes of the studies and 95% confidence intervals (95%CIs) for the differences in means between outcomes were calculated using means and standard deviations (SDs) for each independent study. Cohen’s d and its corresponding 95%CI were calculated using those means and SDs.

Predefined subgroup analyses based on follow-up periods were conducted to detect potential sources of heterogeneity among the studies. Additionally, a post hoc analysis was conducted because two studies reporting HDL-C and cholesterol outcomes showed significant baseline differences between the intervention and control groups. In this analysis, we repeated the meta-analysis excluding these studies to assess whether the overall effect remained consistent.

Due to the small number of studies included in the meta-analysis, the Q test could not be used to test for the variability in TEs attributable to heterogeneity rather than chance. Instead, we used the *I*^2^ (%) statistic, a function of the Q test (*I*^2^ = 100%(Q − *df*)/Q) that does not depend on the number of included studies. The *I*^2^ statistic is interpreted as low (25%), moderate (50%), or high (75%) [28].

### 2.6. Publication Bias

The Doi plot was used to graphically evaluate publication bias. The Doi plot not only enhances the visualization of the asymmetry (where no asymmetry indicates an absence of publication bias) but also allows for quantifying the asymmetry through the Luis–Furuya–Kanamori (LFK) index. LFK index values within ±1 suggest no asymmetry, LFK index values exceeding ±1 but within ±2 indicate minor asymmetry, and LFK index values exceeding ±2 denote major asymmetry [29].

## 3. Results

Of the 494 citations identified through the electronic search, 8 articles were finally included in the systematic review. Details of the literature search process for obtaining these articles are presented in Figure 1.

### 3.1. Study and Intervention Description

As shown in Table 1, the included studies were conducted between 2001 and 2021, with the majority taking place in Saudi Arabia (SA) (n = 4). Half of the studies focused on patients with prediabetes [30,31,32,33], while the others examined patients with diabetes [34,35,36,37]. Most interventions were conducted in primary healthcare centers (n = 5).

The majority of the studies utilized randomized controlled trials (RCTs) (n = 5) followed by quasi-experimental pre–post designs (n = 3). In six studies, the interventions were conducted by healthcare providers in collaboration with dietitians (n = 6), while two studies were solely conducted by dietitians.

The types of interventions varied across studies: three employed individual consultation, two used group-based education interventions, and three studies combined both approaches. Among the group-based education interventions, one focused on healthcare providers rather than patients. These educational interventions covered a range of topics, including the risk of developing diabetes, diabetes management, healthy lifestyle promotion, diabetes self-care practices (such as medication management), dietary management, weight reduction, regular medical check-ups, and physical activity. Some studies (n = 2) also incorporated reminders, personal meetings, phone calls, or messages to enhance intervention adherence. The comparison groups, on the other hand, typically received standard lifestyle advice or non-personalized counseling from healthcare providers.

A total of 15 outcomes were reported across eight studies, representing the range of outcomes reported in the available literature. These outcomes were categorized into two groups: objective measures (e.g., laboratory measures and physical examinations) were examined in seven studies, while subjective measures were assessed in one study. In terms of methodological quality, the overall assessment indicated an intermediate level of quality that ranged from 12 to 24 points out of a possible 28, with an average of 20 (±1.4) points. Notably, only one study reported blinding of those measuring the main outcomes [32].

### 3.2. Meta-Analysis

For the meta-analysis, six out of the eight studies were included. One study was excluded due to its one-group pre–post design without a comparison group and another was excluded for not reporting any objective measures, thus making comparisons unfeasible. Additionally, the latter study was the only one evaluating subjective measures, so no pooled effect could be calculated. To obtain meaningful pooled effects, only objective measures reported in at least five studies were included in the meta-analysis. Due to varying follow-up periods, subgroup analyses were performed based on the time points reported in the studies, resulting in more than one subgroup for most studies.

#### 3.2.1. HbA1c

Five studies were included in the meta-analysis for HbA1c, with 717 participants in the intervention group and 522 in the control group (Figure 2). The overall effect of the intervention compared to usual care was significant: −0.41 (95% CI −0.67, −0.16). The Doi plot indicated considerable asymmetry (LFK index = −1.45). Subgroup analysis revealed that the most significant difference occurred after 6 months: −0.54 (95% CI −0.95, −0.13). However, significant heterogeneity was identified among the studies (*p* < 0.01, *I*^2^ = 88%).

#### 3.2.2. Body Mass Index (BMI)

Five studies were included in the meta-analysis for BMI, with 1,082 participants in the intervention group and 743 in the control group (Figure 3). The overall effect of the intervention compared to usual care was significant: −0.28 (95% CI −0.36, −0.19). The Doi plot indicated symmetrical data (LFK index = 0.9). Subgroup analysis showed the largest effect at 12 months follow-up: −0.43 (95% CI −0.59, −0.26). Significant differences were also found at 6 months follow-up: −0.26 (95% CI −0.38, −0.14), while no significant differences were observed at 3 months.

#### 3.2.3. High-Density Lipoprotein Cholesterol (HDL-C)

Six studies were included in the meta-analysis for HDL-C, with 999 participants in the intervention group and 791 in the control group (Figure 4). The overall effect of the intervention compared to usual care was significant: 0.60 (95% CI 0.36, 0.85). The Doi plot indicated symmetrical data (LFK index = 0.20). Significant effects were found across all follow-up periods, with the largest effect observed at 6 months: 0.64 (95% CI 0.31, 0.98). However, significant heterogeneity was present among the studies (*p* < 0.01, *I*^2^ = 85%). Since two of the six studies showed significant baseline differences in HDL-C levels between the treatment and control groups, we conducted a post hoc analysis. Although the magnitude of the effect was lower, the overall effect of the intervention compared to usual care remained significant (0.41, 95% CI: 0.29 to 0.54).

#### 3.2.4. Fast Glucose Blood (FBG)

Six studies were included in the meta-analysis for FBG, with 1,108 participants in the intervention group and 816 in the control group (Appendix A). The overall effect of the intervention compared to the control was not significant. The Doi plot indicated asymmetrical data (LFK index = 1.44). Subgroup analysis showed a significant effect only at the 12-month follow-up: −0.28 (95% CI −0.42, −0.14). Additionally, significant heterogeneity was identified among the studies (*p* < 0.01, *I*^2^ = 79%).

#### 3.2.5. Cholesterol

Seven studies, comprising 1,108 participants in the intervention group and 716 in the control group, were included in the meta-analysis for cholesterol. The intervention, compared to usual care, did not demonstrate a significant overall effect or show significance across any follow-up period (Appendix B). The Doi plot indicated minor asymmetry (LFK index = −1.04). A post hoc analysis excluding two studies with significant baseline differences in cholesterol levels yielded similar results, with the overall effect and outcomes across time points remaining non-significant.

#### 3.2.6. Systolic and Diastolic Blood Pressure (SBP and DBP, Respectively)

Six studies were included in the meta-analysis for SBP and DBP, with 1021 participants in the intervention group and 768 in the control group (Appendix C and Appendix D, respectively). The overall effect of the intervention compared to usual care was not significant at any follow-up period. The Doi plot for SBP indicated considerable asymmetry (LFK index = 1.45), while the plot for DBP indicated symmetrical data (LFK index = 0.71).

## 4. Discussion

This meta-analysis evaluated evidence from lifestyle interventions involving dietitians that targeted Arab populations across different countries for managing prediabetes and diabetes. The inclusion of both prediabetes and diabetes populations provides a comprehensive perspective on the impact of dietary interventions across the spectrum of glucose metabolism abnormalities. This approach enables an evaluation of the shared outcomes and potential benefits of early intervention for prediabetes and diabetes, highlighting the importance of addressing the continuum of disease progression through tailored lifestyle interventions. The findings indicate that these interventions significantly improve HbA1c and BMI, particularly over longer follow-up periods. However, no significant effects were observed for fasting blood glucose, cholesterol levels, or blood pressure.

This meta-analysis aligns with international guidelines recommending dietitians’ involvement in diabetes management [10,38]. Interventions involving dietitians, either independently or as part of a multidisciplinary team, significantly reduced HbA1c levels, with the larger reduction observed after six months, thus underscoring the importance of sustained intervention [39]. This clinically meaningful decrease of at least 0.5% has been associated with reduced risks of cardiovascular and microvascular complications, emphasizing the critical role of sustained dietary interventions [40]. However, the low certainty of evidence, stemming from heterogeneity and potential bias, limits confidence in these findings. These findings are consistent with previous meta-analyses focusing on medical nutrition therapy (MNT) for prediabetes [41] and diabetes [42], which highlight the efficacy of dietitian involvement in the interventions in achieving clinically significant glycemic control. For example, a meta-analysis that examined the effectiveness of dietitian-delivered nutrition therapy in adults with diabetes demonstrated significant reductions in HbA1c. This analysis, which included 13 studies with a total of 3,338 participants—primarily of white European descent—found that dietitian-delivered MNT achieved clinically and statistically significant reductions in HbA1c, with participants in the intervention group reducing their HbA1c levels four-and-a-half times more than those in the control group [42]. These findings reinforce the need to integrate dietitians in interventions in diabetes care, particularly in culturally unique settings like Arab populations.

In addition to HbA1c, lifestyle interventions significantly reduced BMI, with the largest effect observed at the 12 months. Moreover, we identified consistent findings across studies reporting BMI as an outcome, demonstrating homogeneity in this regard. Similar patterns have been observed in other studies showing that longer intervention durations lead to more significant weight loss [43]. As overweight and obesity are strongly associated with diabetes [44], and weight loss is associated with improved metabolic control [45], weight loss likely mediates the relationship between lifestyle interventions and improved metabolic outcomes. Additionally, BMI may serve as an accessible indicator of the quality and success of dietary interventions [24,41]. However, BMI alone does not capture the complexity of adiposity, such as fat distribution, highlighting the need for future studies to incorporate more nuanced measures of body composition, including waist-to-hip ratio and body fat percentage.

Our results also revealed a significant increase in HDL-C levels among intervention groups. This improvement is likely driven by dietary modifications and weight reduction, as evidenced by the strongest effects observed at six months. Prior research, including studies from Western contexts, has similarly demonstrated that lifestyle changes positively influence HDL-C levels [46,47]. However, conflicting evidence exists; a previous systematic review of Arab populations found no significant changes in HDL-C when dietary advice was provided by non-dietitian professionals [48]. These discrepancies emphasize the critical role of dietitians in delivering tailored and effective nutritional guidance, particularly in populations with culturally specific dietary habits.

Contrary to some previous meta-analyses and systematic reviews, we found no significant improvements in blood pressure or cholesterol levels. For example, Al-Mhanna [49] reported significant improvements in these outcomes with combined lifestyle interventions in high-income countries. The lack of significant findings in our study may be attributed to several factors, including heterogeneity across studies, small sample sizes, which could have limited the statistical power to detect meaningful differences, and the complexity of these outcomes, which are influenced by genetic, lifestyle, and pharmacological factors [50,51]. Additionally, cultural barriers, such as low levels of physical activity among women in Gulf States, may have attenuated the effectiveness of these interventions [52]. Interestingly, subgroup analyses revealed significant reductions in fasting blood glucose at the 12-month follow-up, underscoring the importance of prolonged intervention durations [43]. These findings highlight the need for multifaceted approaches that combine dietary counseling with other components, such as structured physical activity programs and optimized medication use, to achieve broader metabolic improvements.

In accordance with the ADA [10], diabetes management should involve a multidisciplinary team. The studies included in this meta-analysis highlight the importance of a multidisciplinary and multifaceted approach to managing diabetes and prediabetes among Arab populations [34,36,37]. Our findings underscore the need for lifestyle interventions that integrate dietitians as essential members of the healthcare team to provide comprehensive and effective diabetes management. While physicians and nurses often lack comprehensive nutrition education [53,54], dietitians are trained extensively in nutrition education, counseling, and behavior modification strategies. This equips them with the expertise required to deliver effective dietary management and improve patient outcomes [41,42]. Therefore, integrating dietitians into healthcare teams is vital for providing specialized dietary guidance and enhancing diabetes management. These findings suggest that healthcare policymakers should prioritize the inclusion of dietitian-involved interventions in national diabetes management programs. This could involve not only increasing the number of trained dietitians but also ensuring that cultural factors are integrated into healthcare delivery systems to make interventions more effective and sustainable.

The intensity and duration of interventions also emerged as key factors in determining their effectiveness. Studies with more frequent and sustained interactions, such as those by Alfawaz et al. [32] and Wani et al. [33], reported more substantial improvements in clinical outcomes. This underscores the importance of regular contact with healthcare providers to maintain and reinforce lifestyle changes [55], consistent with prior findings that more than 28 sessions lead to significant improvements in BMI and HbA1c [43].

The unique cultural and dietary practices of Arab populations present both challenges and opportunities for diabetes management. Social customs, such as hosting guests with abundant food offerings, and the dietary shifts toward Westernized, calorie-dense foods, complicate adherence to nutritional guidelines. Following the American Diabetes Association’s recommendation for culturally sensitive diabetes self-management education [10,56], our findings underscore the importance of culturally tailored interventions that address these barriers while respecting traditional practices. Thus, personalized approaches, such as those employed in studies by Al-Hamdan et al. [30,31], which adapted intervention delivery through social media, demonstrate the potential of culturally sensitive strategies to improve adherence and outcomes.

Despite the positive findings, this meta-analysis highlights gaps in the current literature. None of the included studies assessed long-term outcomes such as the progression from prediabetes to diabetes, microvascular complications (e.g., nephropathy), macrovascular complications (e.g., cardiovascular events, stroke), or diabetes-related mortality. Additionally, patient-centered outcomes like quality of life, adherence, and self-management behaviors, as well as anthropometric measures (e.g., waist-to-hip ratio, body fat percentage), were notably absent, limiting a comprehensive evaluation of these interventions. Outcomes related to gut health, including microbiome composition, short-chain fatty acid production, and inflammatory markers (e.g., interleukins), were also not reported. Addressing these gaps in future research will allow for a more comprehensive evaluation of the long-term impact of interventions involving dietitians, enhance our understanding of their effects on diabetes management, and better inform clinical practice and policy- and decision-making.

## 5. Limitations

Several limitations should be considered when interpreting these findings. First, the high heterogeneity across studies is a significant concern, as variation in patient characteristics, including age, gender, setting, and diabetes duration, may have influenced the results. The use of random-effects models helps account for this variability, providing more generalized findings across diverse populations. Second, most of the included studies were small-scale and short-term, limiting the ability to assess long-term effectiveness and feasibility. Nevertheless, consistent improvements in key clinical markers, such as HbA1c and BMI, suggest strong potential for these interventions in broader, longer-term applications.

Third, there is a potential for publication bias, particularly regarding the generalizability of findings to non-Gulf Arab populations or those living in diaspora. Confounding factors, such as socioeconomic status and healthcare access, may have further influenced outcomes. To enhance external validity, studies involving Arab populations in other regions with diverse healthcare settings are needed. Finally, the small number of studies included in the meta-analysis may reduce statistical power and generalizability. This limitation reflects the current state of the literature, as no additional studies meeting the inclusion criteria were identified during the comprehensive search.

Despite these limitations, the consistent improvements in HbA1c and BMI underscore the potential effectiveness of dietitian involvement interventions and highlight critical gaps in the research. Future studies should prioritize the inclusion of larger, multi-site samples to better represent diverse populations and healthcare settings. Standardizing outcome measures is also crucial to reducing variability and enabling more robust comparisons across studies. Additionally, expanding the scope of the outcomes to include both long-term measures will provide a more comprehensive understanding of these interventions’ long-term impact and sustainability. Addressing these gaps is essential for advancing the evidence base and informing clinical practice and healthcare policy. Finally, a standardized framework for culturally adapting interventions could help ensure consistency across studies while addressing regional differences.

## 6. Conclusions

This systematic review and meta-analysis highlight the significant benefits of lifestyle interventions involving dietitians for managing prediabetes and diabetes among Arab populations, demonstrating improvements in HbA1c, BMI, and HDL-C levels. The findings emphasize the importance of incorporating dietitians into multidisciplinary healthcare teams and tailoring interventions to the unique dietary practices and cultural norms of Arab populations. Despite limitations, such as study heterogeneity and small sample sizes, these interventions show strong potential for broader and long-term applications. Future research should focus on larger, high-quality studies, expand outcome measures, and explore strategies to enhance sustainability through community engagement and education. These efforts will support the integration of culturally sensitive diabetes management programs and reduce the healthcare burden associated with diabetes in Arab populations.

## Figures and Tables

**Figure 1 nutrients-16-04283-f001:**
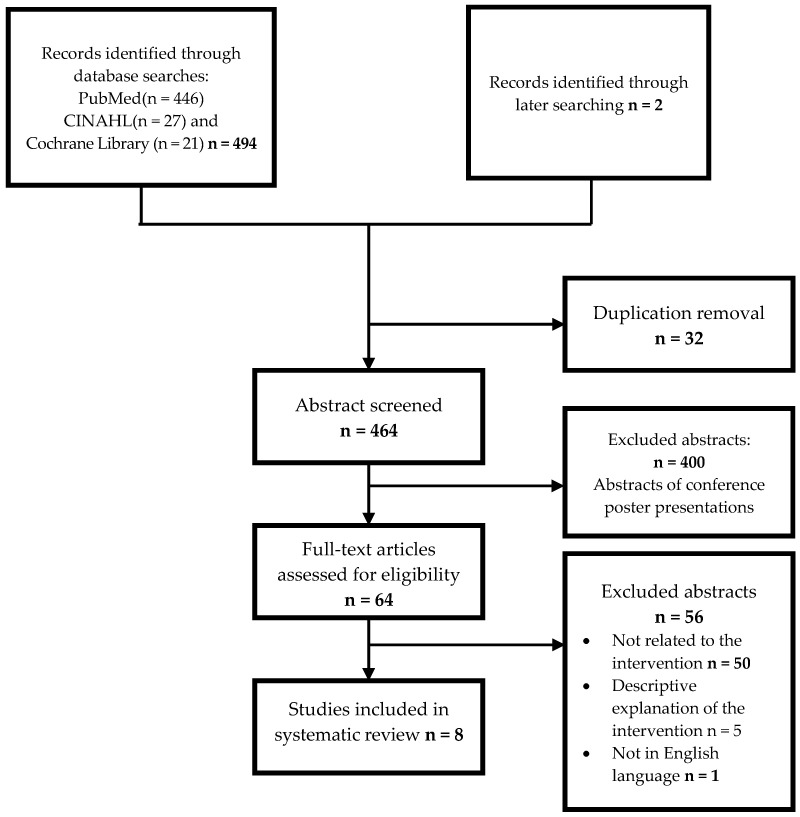
Flow chart of the study selection process.

**Figure 2 nutrients-16-04283-f002:**
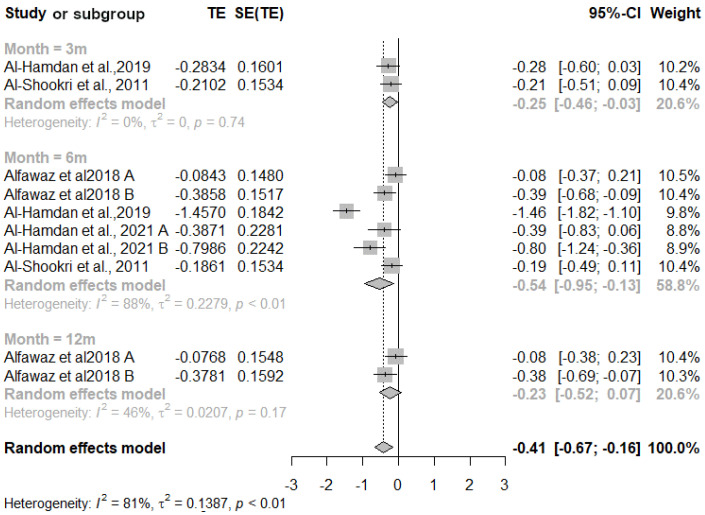
Forest plot diagram of the effect of interventions on hemoglobin A1c (%) levels compared to usual care [30,31,32,35]. Note: CI = confidence interval; TE = treatment effect; SE (TE) = standard error of the treatment.

**Figure 3 nutrients-16-04283-f003:**
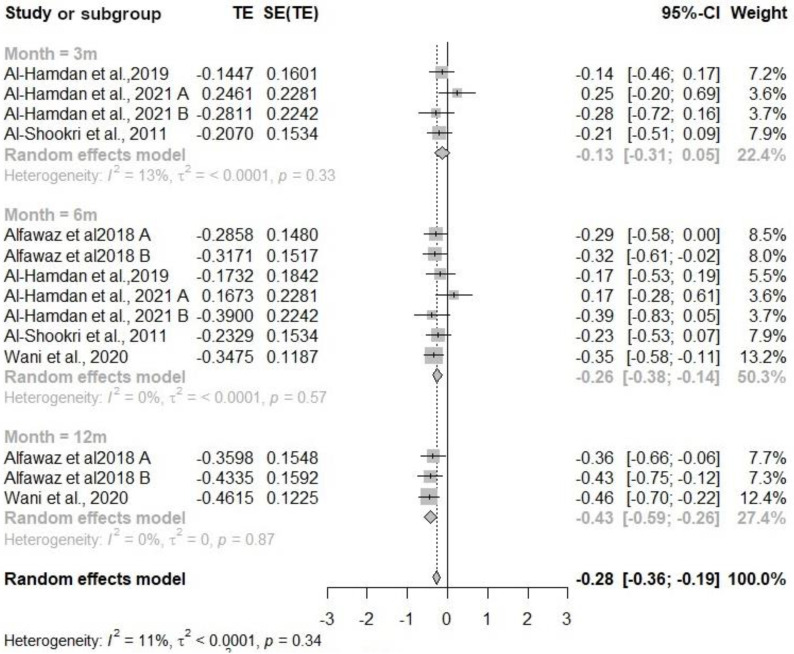
Forest plot diagram of the effect of interventions on body mass index compared to usual care [30,31,32,33,35]. Note: CI = confidence interval; TE = treatment effect; SE (TE) = standard error of the treatment.

**Figure 4 nutrients-16-04283-f004:**
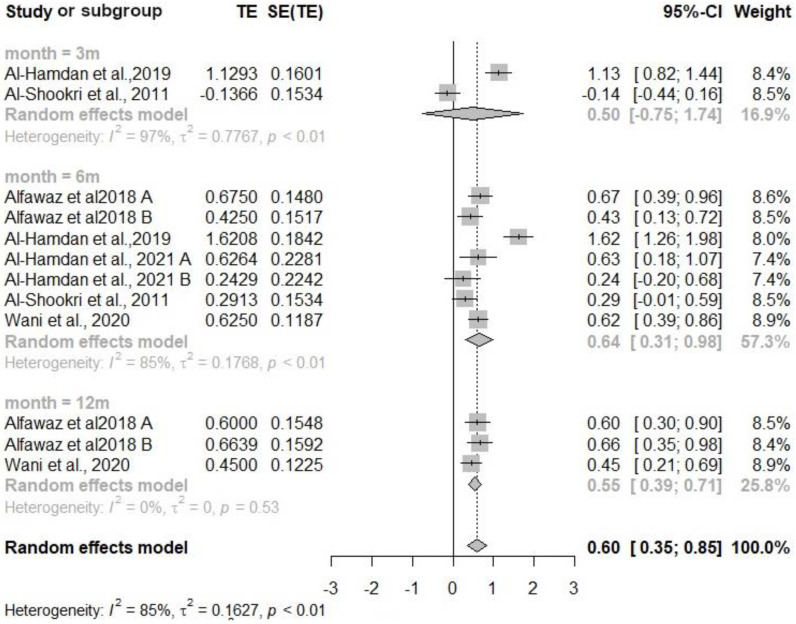
Forest plot diagram of the effect of interventions on high-density lipoprotein level compared to usual care [30,31,32,33,35]. Note: CI = confidence interval; TE = treatment effect; SE (TE) = standard error of the treatment effect.

**Table 1 nutrients-16-04283-t001:** Characteristics of included studies.

Author	Year	Country	Target Group	Sample Size	Mean Age (Years)	Setting	Study Design	Intervention Type	Intervention Content	Intensity of Intervention *	Time of Follow-Up	Intervention Conductor	Quality Assessment
Al-Arifi and Al-Omar [34]	2018	SA	T2DM	174	52.9 ± 14.2	hospital	Quasi-experiment 1 group pre-post	Educational	Medications, nutrition, self- monitoring and management	4 sessions	3, 6, 12 months	Physician dietitians nurse pharmacist diabetic educator	12
Alfawaz et al. [32]	2018	SA	Pre diabetes	294	Int.(A) 43.4 ± 7.8 Int.(B) 42.6 ± 6.9 Cont. 42.3 ± 11.2	Hospital	3-arm RCT	Pharmaceutical and educational	General advice (GA) on lifestyle change; INT. A—intensive lifestyle modification; INT.B-GA + metformin	1 orientation session + seminars of lifestyle modifications every 4 months	6, 12 months	Physicians and dietitians	24
Al-Hamdan et al. [30]	2019	SA	Pre diabetes	190	40.6 ± 9.8	PHC	2-arm RCT	Educational, nutrition and PA advise	one-on-one intensive lifestyle modification	Minimum 6 sessions, NI for time, for 3 months	3, 6 months	Dietitian	21
Al-Hamdan et al. [31]	2021	SA	Pre diabetes	253	Int.(A) 42.9 ± 12.2 Int.(B) 43.7 ± 8.1 Cont. 50.9 ± 7.1	PHC	3-arm RCT	Education, nutrition and PA advise	modification through social media or frontal	6 sessions, twice a month, for 3 months	6 months	Nurses and dietitians	22
Al-Shookri et al. [35]	2011	Oman	T2DM	170	50.7 ± 10.4	Hospital	2-arm RCT	Educational	Nutrition prescription, eating and exercise goals	4 sessions (1 session 1–1.5 h) 2, 3 sessions (30–45 min) for 1–1.5 months, 4 session every (6–12 months)	3, 6 months	Dietitian	20
Reed et al. [36]	2001	UAE	T2DM and T1DM	219	Int. 49.4 ± 11.7 Cont. 53.6 ± 10.9	PHC	Quasi-experiment 2 groups pre-post	Educational	Multifaceted, diabetic nutrition and care, motivational interview	NI sessions, for 12 months	18 months	Physicians, nurses, dietitians	20
Reed et al. [37]	2005	UAE	T2DM and T1DM	738	Int. 53.3 ± 10.9 Cont. 54.1 ± 10.4	PHC	Quasi-experiment 2 groups pre-post	Educational	Multiple modalities, diabetic care, adherence to guidelines.	NI sessions, for 3 months	12 months	Physicians, nurses, dietitians	20
Wani et al. [33]	2020	SA	Pre diabetes	300	Int. 43.10 ± 9.4 Cont. 43.75 ± 10.9	PHC	2-arm RCT	Educational	Intensive modifying lifestyle	4 sessions, 3–4 h each, for 4 months	6,12 months	Dietitian Physician physical therapist	21

Note: UAE, United Arab Emirates; SA, Saudi Arabia; T2DM, type 2 diabetes; T1DM, type 1 diabetes; NI, no information; PHC, primary healthcare; RCT, randomized controlled trial; h, hour; PA, physical activity. * Intensity of intervention includes number of sessions, time for each session, and duration of intervention (as this study presented).

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
