# Peer review of "Effectiveness of Dietitian-Involved Lifestyle Interventions in Diabetes Management Among Arab Populations: A Systematic Review and Meta-Analysis"

_nutrients, 2024, doi:10.3390/nu16244283_

Round 1

Reviewer 1 Report

Comments and Suggestions for Authors

This systematic review and meta-analysis aimed to evaluate the effectiveness of dietitian- involved lifestyle interventions among Arab populations with prediabetes or diabetes. Specifically this study aim to provide valuable insights into the potential of dietitian-involved programs to deliver culturally sensitive and effective diabetes management. Therefore, it is expected that it will be able to provide valuable information to prevent diabetes in Arab populations. However, correction and supplementation of the following matters are required.

1. It seems necessary to explain the background or reason for the results being limited to HbA1c, BMI, HDL, Fasting Glucose, Cholesterol, BP, etc. in the introduction and research method. Additionally, additional explanation on this part should be provided in the discussion.

2. A detailed description should be provided for the excluded papers shown in Figure 1. Although it is a flowchart, it is presented too simply.

3. The number of papers included in the study is too limited. It is need to comment about specific reason or consideration.

4. An explanation of how the characteristics of the study subjects were controlled is required.

 5. The discussion centered on the results is insufficient. While a more detailed discussion is required, it is limited to comparisons with some previous studies.

Author Response

Thank you for your thoughtful comments and for recognizing the potential of our study to provide valuable insights into culturally sensitive diabetes management for Arab populations. We have addressed all the corrections and supplementary recommendations outlined and believe the revisions strengthen the manuscript's quality and relevance.

Comment 1: It seems necessary to explain the background or reason for the results being limited to HbA1c, BMI, HDL, Fasting Glucose, Cholesterol, BP, etc. in the introduction and research method. Additionally, additional explanation on this part should be provided in the discussion.

Response 1: We have clarified the rationale for the outcomes assessed in both the Methods and Results sections, explaining that these outcomes were derived from the reported data in the included studies rather than predefined by our review. In the Discussion, we emphasized the need for future research to explore broader outcomes, such as diabetes progression, complications, and patient-centered measures, to address gaps in the literature (lines: 191-193, 245-246, 417-439).‏

Comment 2:  A detailed description should be provided for the excluded papers shown in Figure 1. Although it is a flowchart, it is presented too simply.

Response 2: Thank you for pointing this out. We have revised Figure 1 to include a detailed breakdown of excluded studies, specifying reasons for exclusion at each stage of the screening process. This revision enhances the clarity and transparency of the manuscript.

Comment 3: The number of papers included in the study is too limited. It is need to comment about specific reason or consideration.

Response 3: The limited number of studies reflects the current state of the literature, as no additional studies meeting our predefined inclusion criteria were identified during our comprehensive search. Similar challenges have been noted in prior meta-analyses (e.g., Rondanelli et al., 2024; Nicola et al., 2024). We have expanded the Limitations section to emphasize the need for larger, multi-site studies with standardized outcome measures and broader scopes to improve generalizability and provide a more comprehensive understanding of dietitian-involved interventions.

Rondanelli, Mariangela, et al. "Does the Ketogenic Diet Mediate Inflammation Markers in Obese and Overweight Adults? A Systematic Review and Meta-Analysis of Randomized Clinical Trials." Nutrients 16.23 (2024): 4002.

‏Nicola, Constantin Alin, et al. "Systematic Review and Meta-Analysis on the Association Between Daily Niacin Intake and Glaucoma." Nutrients 16.21 (2024): 3604.

Comment 4: An explanation of how the characteristics of the study subjects were controlled is required.

 Response 4: We thank the reviewer for this insightful comment. Most studies reported no significant differences between intervention and control groups at baseline for key characteristics. For studies with significant baseline differences (e.g., HDL and cholesterol values), we conducted post hoc analyses excluding these studies, which slightly altered the magnitude of the overall effect. These details have been added to the Methods and Results sections to ensure transparency (lines: 203-207, 289-293, 317-319, 320-322).

 Comment 5: The discussion centered on the results is insufficient. While a more detailed discussion is required, it is limited to comparisons with some previous studies

 Response 5: We have expanded the Discussion section to provide a more comprehensive interpretation of our findings. This includes comparisons with additional meta-analyses on medical nutrition therapy and culturally tailored interventions, highlighting consistencies and discrepancies. We also elaborated on the clinical implications of our findings for diabetes management in Arab populations, emphasizing the importance of culturally sensitive, multidisciplinary approaches involving dietitians (p 10-12).

Reviewer 2 Report

Comments and Suggestions for Authors

I have reviewed the manuscript titled "Effectiveness of Dietitian-Involved Lifestyle Interventions in Diabetes Management Among Arab Populations: A Systematic Review and Meta-Analysis". Overall, the paper presents significant findings in the scientific context of metabolic diseases. Nevertheless, some modifications are needed to improve the overall quality of the manuscript.

-        Please remove the space after “dietitian-“ throughout the manuscript.

-        The References in the main text do not follow MDPI’s guidelines. Please revise them.

-        Line 57: please substitute “to achieving” with “to achieve”

-        Clarify the implications of "low certainty of evidence" in the abstract with a brief explanation.

-        Provide a more detailed rationale for the inclusion of both prediabetes and diabetes populations in the analysis.

-        Figure 1 is missing some data. Please modify it.

-        The limitations are appropriately acknowledged but should also address the potential for publication bias given the generalizability of findings to non-Gulf Arab populations or those living in diaspora and the influence of confounding factors like socioeconomic status or healthcare access on outcomes.

-        Provide practical recommendations for integrating these findings into healthcare policies in Arab countries.

Author Response

Thank you for the positive feedback on our manuscript. Below are our responses and the revisions made to address each comment.

Comment 1: Please remove the space after “dietitian-“ throughout the manuscript

Response 1: We have addressed this concern by consistently.

Comment 2: The References in the main text do not follow MDPI’s guidelines. Please revise them.

 Response 2: Thank you for this comment. We have revised the references in the main text to strictly follow MDPI’s guidelines.

 Comment 3: Line 57: please substitute “to achieving” with “to achieve”

 Response 3: This change has been made as suggested (line 63).

Comment 4: Clarify the implications of "low certainty of evidence" in the t abstract with a brief explanation.

 Response 4: Thank you for pointing this out. We have clarified in the abstract that the "low certainty of evidence" stems from heterogeneity and potential bias across the included studies, which limits confidence in the magnitude of the observed effects (lines: 25-26).

 Comment 5: Provide a more detailed rationale for the inclusion of both prediabetes and diabetes populations in the analysis.

 Response 5: Following your suggestion, we have expanded the rationale for including both prediabetes and diabetes populations in the Introduction and Discussion sections. Specifically, we highlighted that both groups share overlapping pathophysiological mechanisms and benefit from similar lifestyle interventions, making their combined analysis both relevant and informative ( lines: 150-153, 332-337).

 Comment 6Figure 1 is missing some data. Please modify it

 Response 6: Thank you for pointing this out. In response to this and Reviewer 1's similar comment, we have revised Figure 1 to include a detailed breakdown of excluded studies, specifying the reasons for exclusion at each stage of the screening process. This revision enhances the clarity and transparency of the manuscript.

 Comment 7: The limitations are appropriately acknowledged but should also address the potential for publication bias given the generalizability of findings to non-Gulf Arab populations or those living in diaspora and the influence of confounding factors like socioeconomic status or healthcare access on outcomes.

Response 7: We agree with this suggestion. Accordingly, we have added a discussion of these limitations in the Limitations section, specifically addressing potential publication bias, generalizability to non-Gulf Arab populations, and the role of confounding factors such as socioeconomic status and healthcare access in influencing outcomes (lines: 450- 454).

Comment 8: Provide practical recommendations for integrating these findings into healthcare policies in Arab countries.

 Response 8: Thank you for this insightful comment. We have added practical recommendations in the Discussion and Conclusion sections. These include emphasizing the integration of dietitians into national diabetes management programs, promoting culturally tailored interventions, expanding training opportunities, and investing in research to explore effective implementation strategies ( lines:460-466, 470-481).

Round 2

Reviewer 1 Report

Comments and Suggestions for Authors

It seems that this paper is revised relatively well as review points.